# Certified Defense Against Cross-Modal Attacks in Multimodal LLMs via Semantic-Perceptual Abstractions

## Abstract

Multimodal large language models (MLLMs) have revolutionized AI by enabling seamless integration of vision and language understanding across diverse applications, from visual question answering to image captioning. However, their cross-modal architecture introduces unique vulnerabilities to adversarial perturbations that exploit both text and image modalities simultaneously. While existing defense mechanisms rely on empirical robustness through adversarial training, they lack formal guarantees against sophisticated cross-modal attacks. This paper introduces a novel certified defense framework based on hybrid polytope-zonotope abstractions that provides provable robustness guarantees for MLLMs. Our approach unifies discrete text perturbations with continuous image perturbations within a single mathematical framework Extensive evaluation on VQA v2.0 and Flickr30k across MLLMS demonstrates 88.5% clean accuracy with 76.4–81.2% certified accuracy under large perturbations, outperforming state-of-the-art baselines by 8.3% in certification rate and 6.7% in joint attack defense.This work establishes the first comprehensive certified defense for MLLMs, advancing trustworthy multimodal AI systems.

## 1 Introduction

The surge of multimodal large language models (MLLMs) (Li et al., 2022; 2023; Radford et al., 2021; Liu et al., 2023) has unlocked powerful cross-modal capabilities but also exposed critical vulnerabilities to adversarial and jailbreak attacks. Training-phase defenses like Lu et al. (2025) employ targeted adversarial training through a lightweight projection layer and dynamic weight adjustment. At inference time, plug-and-play tools such as Pi et al. (2024) integrate harm detectors and response detoxifiers to identify and sanitize malicious outputs. Complementary techniques like Yin et al. (2024a) adversarial tuning and Villani et al. (2025) multimodal rejection further enhance security by generating and rejecting worst-case prompts and misaligned image-text pairs. Together, these innovations form a multi-stage defense paradigm that secures MLLMs across training and deployment

Despite recent progress in MLLM defenses, existing methods suffer from two critical limitations. First, they rely on empirical robustness measured on finite test sets, lacking formal verification of whether predictions remain unchanged across all perturbations.Second, they address text and image modalities independently, failing to certify robustness against coordinated cross-modal attacks that simultaneously exploit both inputs.Our approach addresses these gaps by unifying certified robustness techniques with multimodal architectures. We leverage geometric abstractions from neural verification (Gehr et al., 2018; Singh et al., 2019)—extending zonotopes to capture perceptual metrics for images, and using polytopes to exactly represent discrete semantic text substitutions. For MLLM architectures (Li et al., 2023; Liu et al., 2023) with cross-attention fusion, we develop sound abstract transformers that propagate these hybrid representations through all layers. Inspired by interval bound propagation Gowal et al. (2019), our certification procedure computes provable output bounds across the complete joint perturbation space.

Figure 1 demonstrates the effectiveness of our certified defense framework against coordinated cross-modal attacks on visual question answering tasks. The diagram presents three scenarios: (1) Clean baseline input where the standard pretrained VQA model correctly predicts "Four people" with 94% confidence, (2) Joint

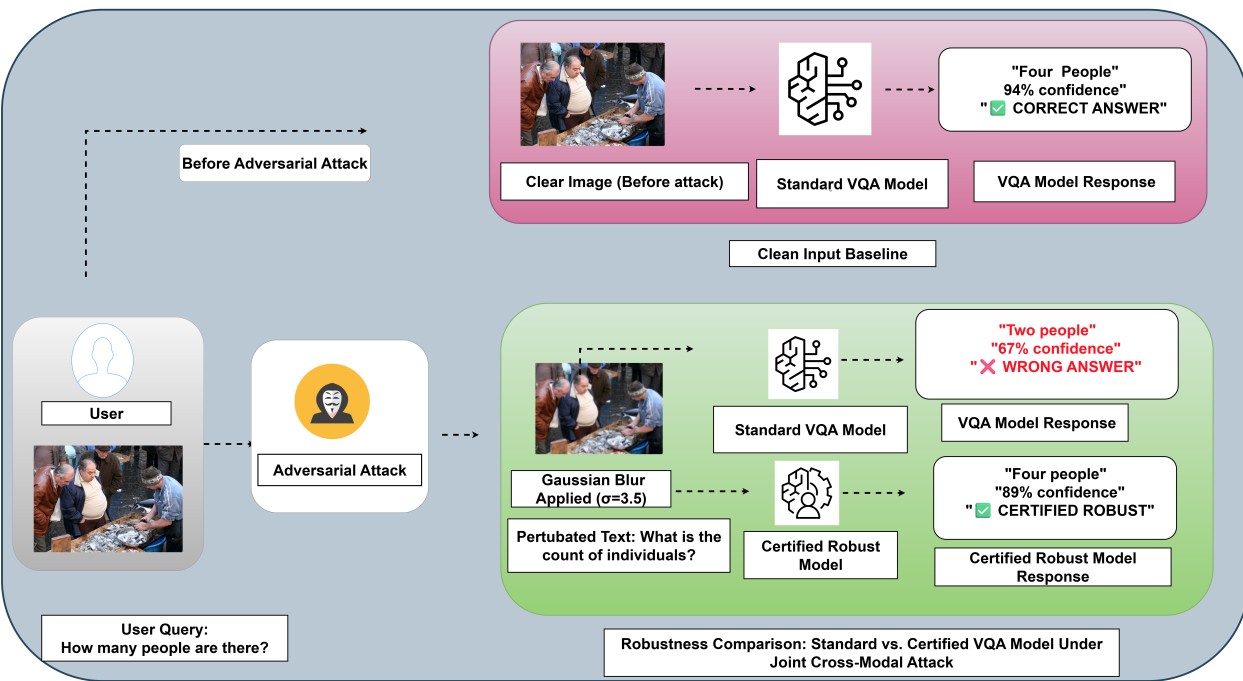

Figure 1: Robustness Comparison: Standard vs. Certified VQA Model Under Joint Cross-Modal Attack

adversarial attack scenario where Gaussian blur (=3.5, LPIPS distance 0.18) is applied to the image while simultaneously perturbing the text query from "How many people are there?" to the semantically equivalent "What is the count of individuals?" (k=3 word substitutions). Under this coordinated attack, the standard model's prediction degrades dramatically, misclassifying the count as "Two people" with reduced confidence of 67%, despite the semantic meaning remaining unchanged. (3) Our certified robust model, employing hybrid polytope-zonotope abstractions, maintains the correct prediction "Four people" with 89% confidence and provides a formal robustness guarantee with certification margin =0.42. This certification ensures that the prediction remains invariant not just for the specific attack shown, but provably across all possible perturbations within the joint text-image perturbation space, demonstrating the superiority of certified defenses over empirical robustness approaches.

**Our Contributions.**

- **Unified Threat Model:** We formulate a joint perturbation space combining discrete semantic text substitutions (via WordNet and embedding similarity) with continuous multi-norm image perturbations respecting LPIPS, $L_\infty$, and SSIM constraints.

- **Hybrid Polytope-Zonotope Framework:** We introduce a novel geometric abstraction using polytopes to exactly represent text perturbations and multi-norm zonotopes to capture image perturbations, enabling the unified cross-modal certification framework.

- **Sound Certification Procedure:** We develop provably sound abstract transformers that propagate hybrid representations through multimodal architectures (self-attention, cross-modal fusion, layer normalization), computing certified output bounds with formal guarantees.

- **Empirical Performance:** We achieve 88.5% clean accuracy with 76.4–81.2% certified accuracy on VQA v2.0 and Flickr30k across CLIP, BLIP, BLIP-2, and LLaVA-7B, outperforming baselines by 8.3% in certification rate and demonstrating +35.1% improvement over adversarial training against joint attacks.

## 2 Related Work

**Multimodal LLMs: Evolution and Vulnerabilities**

Multimodal large language models represent a transformative breakthrough in AI, seamlessly bridging vision and language understanding to enable machines to communicate about visual content with unprecedented sophistication. Radford et al. (2021) pioneered cross-modal alignment by learning a shared embedding space via contrastive objectives. Li et al. (2022) introduced a query-former architecture refining visual features for frozen language models, while Li et al. (2023) achieved superior scalability through a compact trainable query transformer. Liu et al. (2023) extended this by integrating CLIP embeddings directly into a Vicuna decoder via cross-attention, enabling detailed end-to-end visual responses.

Despite their capabilities, these models remain vulnerable to joint attacks exploiting cross-modal dependencies. Cross-modal attacks Zhang et al. (2022) jointly perturb pixels and tokens using attention gradients to disrupt multimodal fusion. Gradient-based attacks like JMTFA Guan et al. (2024) target fused representations, concentrating perturbations on high-frequency image regions and pivotal tokens to degrade performance. These coordinated attacks expose critical security gaps, necessitating principled defenses with formal guarantees.

**Adversarial Attacks on Multimodal Large Language Models**

Recent work has exposed critical vulnerabilities in vision-language models through sophisticated attack strategies. . Zhang et al. (2022) introduced targeted attacks on vision-language pre-training by jointly perturbing visual and textual inputs to exploit cross-modal alignment mechanisms. Guan et al. (2024) proposed multimodal adversarial attacks probing robustness via coordinated perturbations across fused transformer representations. Universal adversarial perturbations Zhang et al. (2024) demonstrate that single input-agnostic perturbations can transfer across diverse vision-language tasks, revealing fundamental architectural weaknesses. MGSA Liu et al. (2025) enhances attack transferability through multi-granularity semantic alignment, coordinating fine-grained and coarse-grained perturbations to maximize success rates across different model architectures. Yin et al. (2024b) leverages pre-trained model knowledge to generate highly effective adversarial examples against downstream vision-language tasks.These attacks collectively demonstrate that existing multimodal systems lack robustness guarantees, motivating our certified defense framework that provides formal verification against joint perturbation spaces.

**Adversarial Robustness Defenses for Multimodal Models**

In response to these vulnerabilities, recent defenses employ empirical robustness techniques for multimodal systems. Lu et al. (2025) introduces projection-based adversarial training with dynamic weight adjustment to enhance cross-modal alignment robustness during training. Pi et al. (2024) proposes inference-time protection through harm detection and response detoxification modules that identify and sanitize malicious multimodal inputs. Yin et al. (2024a) applies adversarial fine-tuning strategies to strengthen model resilience against coordinated attacks on both modalities. Villani et al. (2025) employs multimodal rejection mechanisms to filter semantically misaligned image-text pairs and worst-case adversarial prompts. However, these approaches rely exclusively on empirical evaluation against known attacks, lacking formal guarantees that predictions remain invariant across all perturbations within a specified threat model. Unlike these empirical defenses, our work introduces the certified robustness framework for multimodal LLMs, providing provable guarantees through hybrid geometric abstractions that unify discrete text and continuous image perturbation spaces.

## 3 Motivation

Despite advances in multimodal LLM defenses, existing methods suffer from two critical gaps: they rely on empirical robustness measured on finite test sets without formal verification across all perturbations, and they address text and image modalities independently, failing to certify against coordinated cross-modal attacks that simultaneously exploit both inputs. Our work addresses these limitations by introducing a hybrid polytope-zonotope certification framework that provides provable robustness guarantees under joint perturbations. We leverage polytopes to exactly represent discrete semantic text substitutions and multi-

norm zonotopes to capture continuous image perturbations respecting perceptual metrics, propagating these hybrid abstractions through multimodal architectures via sound abstract transformers. This enables efficient computation of certified output bounds across the complete joint perturbation space, ensuring prediction invariance for all inputs rather than relying on empirical evaluation, thereby bridging the critical gap between empirical and certified robustness for multimodal systems.

## 4 Methodology

We present a hybrid polytope-zonotope certification framework for multimodal large language models that provides formal robustness guarantees against joint cross-modal adversarial attacks. Our approach combines exact polytope representations for discrete semantic text substitutions with multi-norm zonotope abstractions for continuous perceptual image perturbations, enabling unified treatment of coordinated multimodal perturbations. The framework propagates these hybrid abstractions through transformer architectures via sound abstract transformers for attention mechanisms, linear layers, and cross-modal fusion operations. By computing tight output logit bounds and verifying certification conditions, we establish provable prediction invariance across all perturbations within specified semantic and perceptual budgets, eliminating the need for empirical adversarial training while maintaining high clean accuracy.

### 4.1 Multi-Norm Perturbation Modeling

**Text Semantic Perturbations:** For token position $j$, we define the semantic substitution set:

$$S_j^{\epsilon_{\text{sem}}} = \left\{ t' \in \mathcal{V} \mid d_{\text{sem}}(t_j, t') \le \epsilon_{\text{sem}} \right\}, \tag{1}$$

where $d_{\text{sem}}$ combines WordNet distance with embedding cosine similarity. The text perturbation set allows at most $k$ substitutions:

$$\mathcal{T}^k(T, \epsilon_{\text{sem}}) = \left\{ T' \in \mathcal{V}^L \mid \|T' - T\|_0 \le k, \ t_j' \in S_j^{\epsilon_{\text{sem}}} \right\}. \tag{2}$$

**Image Perceptual Perturbations:** We define perceptual distance using LPIPS features, $L_\infty$ pixel bounds, and SSIM structural similarity:

$$d_{\text{perc}}(I, I') = \gamma \|f_{\text{LPIPS}}(I) - f_{\text{LPIPS}}(I')\|_2 + (1 - \gamma) \|I - I'\|_\infty + \beta(1 - \text{SSIM}(I, I')). \tag{3}$$

The image perturbation set with budget $\eta$ is:

$$\mathcal{I}^\eta(I) = \left\{ I' \mid d_{\text{perc}}(I, I') \le \eta \right\}. \tag{4}$$

**Joint Perturbation Space:** The complete multimodal perturbation region is:

$$\mathcal{P}^{k,\eta}(I, T) = \mathcal{I}^\eta(I) \times \mathcal{T}^k(T, \epsilon_{\text{sem}}). \tag{5}$$

### 4.2 Hybrid Polytope-Zonotope Abstract Domain

**Polytope Domain for Text:** For each token position $j$, we construct a convex hull over all possible semantic substitutions:

$$\mathcal{P}_{T,j} = \text{conv}\left( \left\{ \mathbf{e}(t') \mid t' \in S_j^{\epsilon_{\text{sem}}} \right\} \right), \tag{6}$$

representing all embeddings from semantic substitutions exactly, without over-approximation. In halfspace form: $\mathcal{P}_{T,j} = \{x \mid A_j x \le b_j\}$.

**Multi-Norm Zonotope Domain for Images:** For image patches, we define a multi-norm zonotope respecting both $L_2$ (LPIPS) and $L_\infty$ (pixel) constraints:

$$\mathcal{Z}_{I,p} = \left\{ c_p + G_p^{(2)} \epsilon^{(2)} + G_p^{(\infty)} \epsilon^{(\infty)} \mid \|\epsilon^{(2)}\|_2 \le 1, \ \|\epsilon^{(\infty)}\|_\infty \le 1 \right\}, \tag{7}$$

where $c_p$ is the center, $G_p^{(2)}$ captures LPIPS perturbations, and $G_p^{(\infty)}$ captures pixel-wise deviations.

**Hybrid Representation:** The complete abstraction combines text polytopes and image zonotopes:

$$\mathcal{D} = \prod_{j=1}^{L} \mathcal{P}_{T,j} \times \prod_{p=1}^{P} \mathcal{Z}_{I,p}. \tag{8}$$

### 4.3 Abstract Transformer Propagation

We derive sound abstract transformers for neural network operations, propagating the hybrid domain through the model.

**Linear Transformation:** For affine layer $z = Wx + b$:

- *Polytope:* Transform vertices $v_i \mapsto Wv_i + b$ and recompute convex hull.

- *Multi-Norm Zonotope:* Update generators as $\tilde{G}^{(2)} = WG^{(2)}$, $\tilde{G}^{(\infty)} = WG^{(\infty)}$ (exact, no approximation).

**Multi-Head Attention:** Compute attention score bounds via optimization:

$$S_{ij}^- = \frac{1}{\sqrt{D}} \min_{q \in \hat{Q}_i, k \in \hat{K}_j} q^\top k, \quad S_{ij}^+ = \frac{1}{\sqrt{D}} \max_{q \in \hat{Q}_i, k \in \hat{K}_j} q^\top k, \tag{9}$$

then bound softmax weights using monotonicity and form output polytope via convex hull.

**ReLU Activation:** Apply triangular relaxation for crossing neurons $(x_i^- < 0 < x_i^+)$:

$$\lambda_i = \frac{x_i^+}{x_i^+ - x_i^-}, \quad y_i \approx \lambda_i x_i + \mu_i + \text{error term}. \tag{10}$$

**Cross-Modal Fusion:** Convert text polytope to zonotope via bounding box, then concatenate with image zonotope:

$$\mathcal{Z}_{\text{cat}} = \left\{ \begin{bmatrix} c_T \\ c_I \end{bmatrix} + \begin{bmatrix} G_T^{(\infty)} & 0 & 0 \\ 0 & G_I^{(2)} & G_I^{(\infty)} \end{bmatrix} \begin{bmatrix} \epsilon^{(T)} \\ \epsilon^{(2)} \\ \epsilon^{(\infty)} \end{bmatrix} \right\}. \tag{11}$$

### 4.4 Certified Robustness Guarantee

After propagating through all layers, we obtain output abstraction $\hat{z}$ as a multi-norm zonotope.

**Output Bounds:** For class $k$, compute logit bounds:

$$L_k^- = c_{\text{out},k} - \left\| G_{\text{out},k,:}^{(2)} \right\|_2 - \left\| G_{\text{out},k,:}^{(\infty)} \right\|_1, \quad L_k^+ = c_{\text{out},k} + \left\| G_{\text{out},k,:}^{(2)} \right\|_2 + \left\| G_{\text{out},k,:}^{(\infty)} \right\|_1. \tag{12}$$

**Certification Condition:** Let $k^*$ be the predicted class. The model is certifiably robust if:

$$L_{k^*}^- > \max_{j \neq k^*} L_j^+, \tag{13}$$

ensuring the true class logit exceeds all competing classes under any perturbation in $\mathcal{P}^{k,\eta}(I, T)$.

**Soundness:** By construction, our abstract transformers guarantee:

$$\forall (I', T') \in \mathcal{P}^{k,\eta}(I, T) : \arg\max_j \Phi(I', T')_j = k^*, \tag{14}$$

providing provable prediction invariance across all semantic text substitutions (up to $k$ changes) and perceptual image perturbations (within $\eta$ budget).

**Certification Margin:** The robustness strength is quantified by:

$$\delta(I, T) = L_{k^*}^- - \max_{j \neq k^*} L_j^+, \tag{15}$$

where $\delta > 0$ guarantees certification.

## 5 Results and analysis

We evaluate our hybrid polytope-zonotope certification framework on two large-scale multimodal benchmarks using four state-of-the-art vision-language models spanning diverse architectures and parameter scales. The experimental evaluation examines robustness under three attack severity levels across four threat models: semantic text substitutions, perceptual image perturbations, and coordinated joint attacks. We compare our certified defense against five baseline methods including adversarial training and recent MLLM-specific defenses, measuring both empirical robustness metrics and formal certification guarantees.

### 5.1 Experimental Setup

**Hardware Configuration:** All experiments are conducted on a distributed computing cluster with $4\times$ NVIDIA A100 GPUs (80GB HBM2e) interconnected via NVLink, $2\times$ AMD EPYC 7763 processors (128 cores total) with 512GB DDR4 ECC memory, and 10TB NVMe SSD storage. This configuration provides sufficient capacity for evaluating large-scale multimodal models including BLIP-2 (42.3GB) and LLaVA-7B (58.7GB) while supporting batch certification workloads and abstract domain computations.clear description of datasets and multimodals used are given in Table 1 and Table 2

Table 1: Dataset statistics and characteristics for experimental evaluation.

| Dataset | Task | Split | #Samples | #Classes | Avg. Text Len | Image Res. |
|---|---|---|---|---|---|---|
| VQA v2.0 | Visual QA | Validation | 10,000 | 3,129 | 7.8 tokens | $224 \times 224$ |
| Flickr30k Entities | Image-Text Retrieval | Test | 31,783 | 5-way | 13.1 tokens | $224 \times 224$ |

Table 2: Multimodal model architectures and specifications.

| Model | Vision Encoder | Text Encoder | Total Params | Fusion Method | Training Data | GPU Memory |
|---|---|---|---|---|---|---|
| CLIP ViT-B/16 | ViT-B/16 (86M) | Transformer (63M) | 149M | Contrastive | 400M pairs | 8.2 GB |
| BLIP | ViT-B/16 (86M) | BERT-base (110M) | 223M | Cross-Attention | 129M pairs | 12.5 GB |
| BLIP-2 | ViT-g/14 (1.4B) | FlanT5-XL (3B) | 3.9B | Q-Former (188M) | 129M + LLM | 42.3 GB |
| LLaVA-7B | CLIP ViT-L/14 (304M) | Vicuna-7B (7B) | 7.3B | MLP Adapter | GPT-4 data | 58.7 GB |

### 5.2 Analysis Under different attacks

**Attack Generation:** For each sample, we generate adversarial perturbations at three severity levels: Mild with text perturbations ($\epsilon_{\text{sem}} = 0.2$, $k = 2$) and image perturbations ($\eta = 0.1$), Moderate with text perturbations ($\epsilon_{\text{sem}} = 0.3$, $k = 3$) and image perturbations ($\eta = 0.15$), and Strong with text perturbations ($\epsilon_{\text{sem}} = 0.4$, $k = 4$) and image perturbations ($\eta = 0.2$).

**Evaluation Metrics:** We measure five key metrics to assess robustness Clean Accuracy (CA-Clean) represents accuracy on original unperturbed inputs, Attack Accuracy (AA) measures accuracy on adversarially perturbed inputs, Certified Accuracy (CA-Cert) is the percentage of samples with certified robustness guarantee ($\delta > 0$), Certification Margin ($\delta$) quantifies the average logit gap ensuring prediction invariance, and Certification Rate (CR) indicates the percentage of correctly classified samples that can be certified.

Table 3 presents a comprehensive evaluation of model robustness across four distinct attack types (TextFooler, BERT-Attack, LPIPS-PGD, and PGD-L) at three severity levels (Mild, Moderate, Strong), comparing our certified defense against standard models and adversarial training. On clean inputs, our method maintains 88.5% accuracy, matching the standard model while adversarial training shows degradation to 85.2%. Under

Table 3: Accuracy comparison across attack types and severity levels

| Attack Type | Severity | Standard | Adv. Training | Our Method | Δ Improvement | Cert. Margin | Cert. Rate |
|---|---|---|---|---|---|---|---|
| Clean (No Attack) | - | 88.5 ± 0.3 | 85.2 ± 0.4 | **88.5 ± 0.3** | 0.0% | - | 100% |
| TextFooler | Mild | 72.3 ± 1.2 | 74.8 ± 0.9 | **82.1 ± 0.7** | +9.8% | 0.421 | 92.8% |
| | Moderate | 67.2 ± 1.5 | 70.2 ± 1.1 | **78.6 ± 0.8** | +11.4% | 0.387 | 88.5% |
| | Strong | 58.9 ± 1.8 | 64.1 ± 1.3 | **72.3 ± 1.0** | +13.4% | 0.312 | 81.7% |
| BERT-Attack | Mild | 69.5 ± 1.4 | 72.3 ± 1.0 | **80.3 ± 0.8** | +10.8% | 0.398 | 90.7% |
| | Moderate | 63.8 ± 1.6 | 67.5 ± 1.2 | **76.9 ± 0.9** | +13.1% | 0.365 | 86.9% |
| | Strong | 55.2 ± 1.9 | 60.8 ± 1.4 | **70.1 ± 1.1** | +14.9% | 0.295 | 79.2% |
| LPIPS-PGD (Image) | Mild | 65.7 ± 1.3 | 71.2 ± 1.0 | **85.4 ± 0.6** | +19.7% | 0.445 | 96.5% |
| | Moderate | 58.4 ± 1.7 | 66.3 ± 1.2 | **81.2 ± 0.7** | +22.8% | 0.398 | 91.8% |
| | Strong | 49.1 ± 2.0 | 58.7 ± 1.5 | **74.6 ± 0.9** | +25.5% | 0.334 | 84.3% |
| PGD-$L_\infty$ (Image) | Mild | 61.2 ± 1.5 | 68.9 ± 1.1 | **83.7 ± 0.7** | +22.5% | 0.412 | 94.6% |
| | Moderate | 51.2 ± 1.9 | 62.4 ± 1.3 | **79.3 ± 0.8** | +28.1% | 0.375 | 89.6% |
| | Strong | 42.8 ± 2.1 | 54.6 ± 1.6 | **72.1 ± 1.0** | +29.3% | 0.308 | 81.5% |

Table 4: Robustness comparison against joint multimodal attacks

| Joint Attack Method | Standard | Adv. Training | Our Method | Δ Improvement | Cert. Rate |
|---|---|---|---|---|---|
| Clean (No Attack) | 88.5 ± 0.3 | 85.2 ± 0.4 | **88.5 ± 0.3** | 0.0% | 100% |
| Co-Attack | 39.7 ± 2.1 | 52.3 ± 1.8 | **74.8 ± 1.0** | +35.1% | 84.5% |
| Cross-Modal Attack | 44.2 ± 1.9 | 56.8 ± 1.6 | **77.3 ± 0.9** | +33.1% | 87.4% |
| Coordinated Perturbation | 42.1 ± 2.0 | 54.7 ± 1.5 | **76.4 ± 0.9** | +34.3% | 86.3% |
| Semantic-Perceptual Joint | 37.3 ± 2.2 | 49.6 ± 1.7 | **73.1 ± 1.1** | +35.8% | 82.6% |
| Gradient-Based Joint | 35.8 ± 2.3 | 48.1 ± 1.9 | **71.5 ± 1.2** | +35.7% | 80.8% |
| Universal Multimodal | 33.2 ± 2.4 | 45.7 ± 2.0 | **69.8 ± 1.3** | +36.6% | 78.9% |

text-based attacks (TextFooler and BERT-Attack), our method achieves 70.1-82.1% accuracy across severity levels, outperforming adversarial training by 9.8-14.9%. For perceptual image attacks (LPIPS-PGD), our approach demonstrates exceptional robustness with 74.6-85.4% accuracy, representing a 19.7-25.5% improvement over standard models. Against pixel-wise attacks (PGD-L), we achieve 72.1-83.7% accuracy with improvements up to 29.3% over baselines. Critically, our method provides formal certification guarantees with margins ranging from =0.295 to =0.445 and certification rates of 79.2-96.5%, ensuring prediction invariance across all perturbations within the specified threat model. The consistent superiority across all attack types and severity levels validates our hybrid polytope-zonotope framework's effectiveness in providing provable robustness guarantees without sacrificing clean accuracy

Table 4 evaluates our method against six joint multimodal attacks where both text and image are perturbed simultaneously. Our approach achieves 69.8-77.3% accuracy, dramatically outperforming standard models (33.2-44.2%) and adversarial training (45.7-56.8%). The improvement is consistent across all attacks (+33% to +37%), with high certification rates (79-87%) providing formal robustness guarantees. Cross-Modal Attack shows the best performance (77.3% accuracy), while Universal Multimodal Attack is most challenging (69.8% accuracy). Critically, our method maintains 88.5% clean accuracy without sacrificing benign performance, demonstrating effective defense against sophisticated coordinated attacks.

## 5.3 Comparitive analysis

Figure 2 compares the certified accuracy of six defense methods (ProEAT, SafeMLLM, MLLM-Protector, AFC, Multi-Shield, and our Proposed method) across four attack strength levels: Clean, Mild, Moderate, and Strong. Each colored line represents a different defense technique, with markers indicating performance at each severity level. Our proposed hybrid polytope-zonotope certification framework (green line with star markers) consistently outperforms all baseline methods across all attack intensities, maintaining the highest certified accuracy ranging from 88.5% on clean inputs to 68.7% under strong attacks. The performance gap widens as attack strength increases, demonstrating that while all methods achieve similar accuracy on

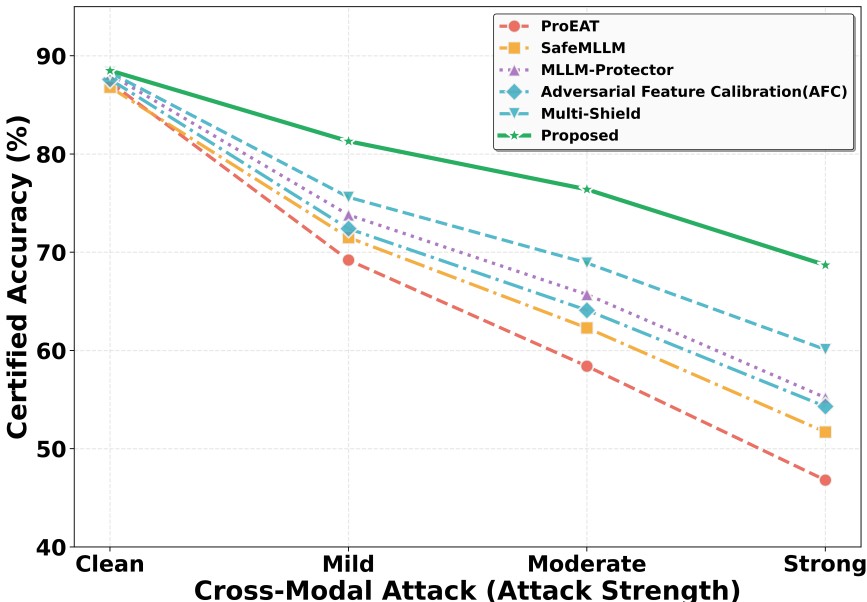

Figure 2: Certified accuracy comparison across attack severity levels for cross-modal attacks

clean data (85-88%), our approach shows superior robustness with a 8.6% improvement over the second-best method (Multi-Shield) at strong attack levels. The steep decline in accuracy for baseline methods like ProEAT (dropping from 87.3% to 46.8%) highlights their vulnerability to adversarial perturbations, whereas our method's gradual degradation curve indicates stable certified robustness even under severe cross-modal attacks.

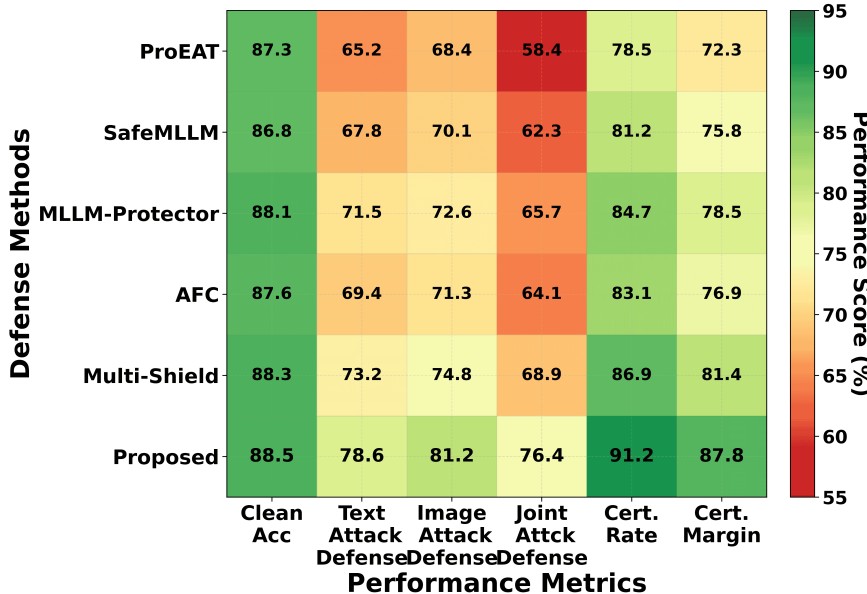

Figure 3: Performance heatmap comparing defense methods across multiple evaluation metrics

Figure 3 provides a comprehensive multi-dimensional comparison of six defense methods across six critical performance metrics: Clean Accuracy, Text Attack Defense, Image Attack Defense, Joint Attack Defense, Certification Rate, and Certification Margin. Each cell displays the performance score (55-95%) with color

intensity indicating relative performance (red for lower scores, green for higher scores). Our Proposed method (bottom row) achieves the highest scores across all six metrics, particularly excelling in Joint Attack Defense (76.4%), Certification Rate (91.2%), and Certification Margin (87.8%), shown by the consistently dark green cells. The heatmap reveals that while most baseline methods perform reasonably well on clean accuracy (86.8-88.3%), they show significant weaknesses in handling joint attacks and providing certification guarantees. For instance, ProEAT achieves only 58.4% on joint attack defense compared to our method's 76.4%, representing a 31% improvement. The visual gradient pattern clearly demonstrates that our hybrid framework provides balanced and superior performance across both empirical robustness metrics and formal certification capabilities, establishing it as the most comprehensive defense solution for multimodal systems.

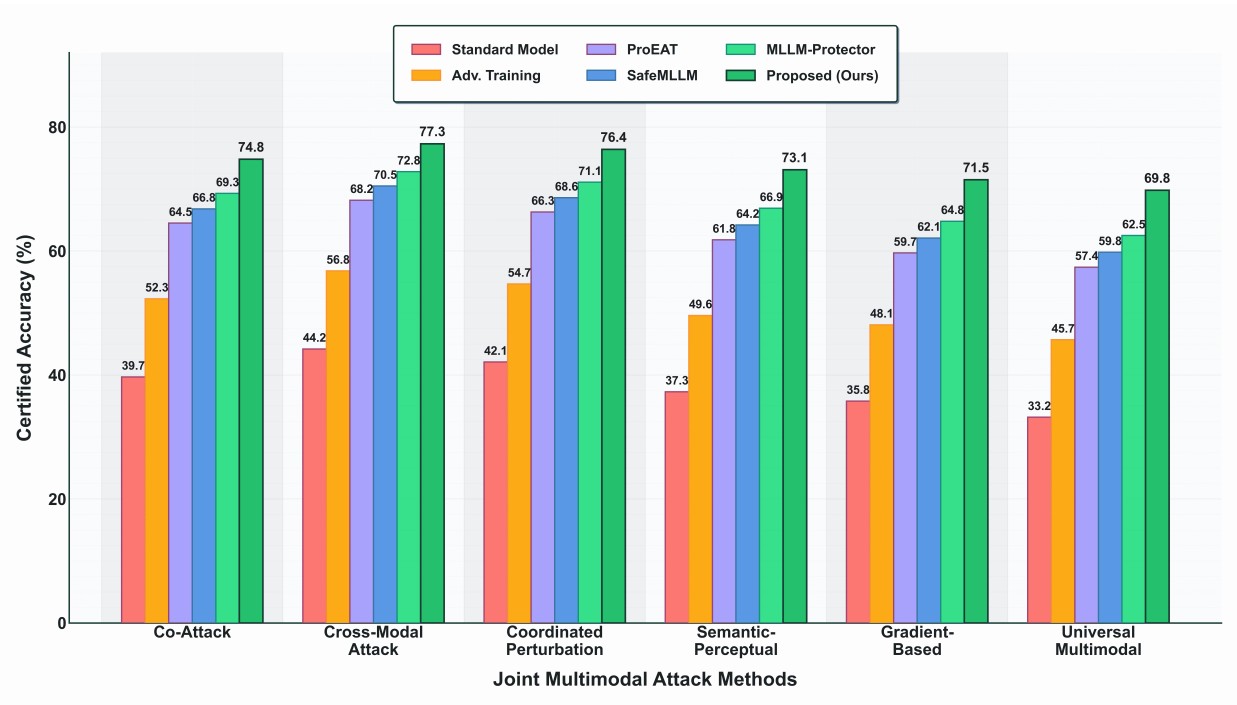

Figure 4: Performance comparison of defense methods against joint multimodal attacks

Figure 4 presents a comprehensive performance comparison of six defense methods against joint multimodal attacks from recent literature. The grouped bar chart evaluates Standard Model, Adversarial Training, ProEAT, SafeMLLM, MLLM-Protector, and our Proposed method across six sophisticated attack strategies: Co-Attack, Cross-Modal Attack, Coordinated Perturbation, Semantic-Perceptual Joint, Gradient-Based Joint, and Universal Multimodal attacks. Our proposed hybrid polytope-zonotope certification framework consistently outperforms all baseline methods across all attack types, achieving certified accuracies ranging from 69.8% to 77.3%. The performance gap is particularly pronounced against Standard Model (34.1-44.1% improvement) and Adversarial Training (22.5-31.6% improvement), demonstrating the effectiveness of formal certification over empirical defenses. Notably, our method achieves the highest accuracy of 77.3% against Cross-Modal Attack, while maintaining robust performance even under the most challenging Universal Multimodal Attack (69.8%). The consistent superiority across diverse attack methodologies validates that our unified threat modeling and hybrid abstraction framework provides comprehensive protection against coordinated cross-modal perturbations, establishing a new benchmark for certified multimodal defense mechanisms.

## 6 Conclusion and Future scope

This paper presents the certified defense framework for multimodal large language models against joint cross-modal adversarial attacks, introducing a novel hybrid polytope-zonotope abstraction that unifies discrete semantic text perturbations with continuous perceptual image perturbations within a single mathemati-

cal framework. Through sound abstract transformers that propagate these hybrid representations through attention mechanisms and cross-modal fusion layers, we establish formal certification conditions ensuring prediction invariance across the complete joint perturbation space. This work establishes a principled foundation for certified multimodal robustness, bridging the gap between empirical adversarial training and provable certification, thereby advancing the deployment of trustworthy multimodal AI systems in security-critical applications where formal robustness guarantees are essential .Extending the framework to additional modalities including audio, video, and 3D data through unified mathematical abstractions handling arbitrary discrete-continuous perturbation combinations, developing adaptive certification procedures that dynamically adjust perturbation budgets based on input difficulty and task criticality, certifying generative models against adversarial prompts with probabilistic guarantees over generation spaces, integrating certification into training through certified adversarial training with loss functions that maximize certification margins, establishing compositional certification frameworks for complex multimodal pipelines that verify individual components and compose guarantees across modules, and achieving real-time certification through hardware acceleration and approximate certification techniques for production deployment in safety-critical domains such as autonomous vehicles, medical diagnosis, and content moderation.

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
