# OpenReview forum: "Certified Defense Against Cross-Modal Attacks in Multimodal LLMs via Semantic-Perceptual Abstractions"
_TMLR — Rejected by TMLR_

### Review · Reviewer_fVtP · 2025-11-03

**Summary Of Contributions:**

Dear authors,

Below you will find my review comments regarding different aspects of your submission. For transparency, I note that **I am not an expert** in the field of multimodal large language models (MLLMs) or vision-language models (VLMs) and their evaluation in terms of robustness, as I have not published in this area. This means feel free to point out if I've misunderstood anything. In this case, I'm willing to reevaluate my review comments.

Looking forward to a productive discussion with you!

**Contributions:**
- The paper identifies certified robustness for MLLMs as an underexplored but highly relevant topic (see Sections 1 and 2).
- A hybrid polytope-zonotope framework with text (see Section 4.1) and image perturbations (see Section 4.2), defining a hybrid domain to be propagated through abstract transformers (see Section 4.3), equips MLLMs with certified robustness (see Section 4.4)
- An extensive evaluation demonstrates the superiority of the proposed framework (see Section 5).

**Strengths:**
- The proposed hybrid polytope-zonotope certification framework for MLLMs offers superior performance gains compared to standard training, adversarial training, and other defense methods.
- The empirical evaluation is quite extensive, as it covers four multimodal models, two large-scale multimodal tasks, five other defense methods as competitors, and multiple attack types with varying levels of severity.
- The presentation of the quantitative performance results in the form of tables and plots is concise, allowing readers to easily assess the performance gains of the proposed framework.
- The graphical abstract in Fig. 1 is a clear and straightforward illustration of the paper's objectives and setting.

**Weakness:**
- The explanations in Section 4 are very brief, particularly regarding its mathematical definitions, which makes understanding very difficult for non-experts (such as me) in the field.
- The related work could be extended to include other relevant works regarding certified and empirical defense for unimodal and multimodal models, e.g., see Section 2 in [I] and Section 2 in [II]. For example, it would be interesting to highlight the differences between your framework and MMCert or randomized ablation [I].
- No codebase has been provided for inspection (neither as supplementary material nor via an anonymous link), and the authors do not state whether any code will be released upon acceptance.

**Additional Comments:**

**Minor Issues:**
- Use \cite when the authors are the grammatical subject or object of the sentence, and use \citep when the citation is not part of the sentence grammar, but just a reference in parentheses.
- In the abstract, a dot is missing after "a single mathematical framework."
- In the abstract, a space is missing after "in joint attack defense."
- In the second paragraph of Section 1, a space is missing after "across all perturbations" and "exploit both inputs."
- In the second paragraph of Section 2, a duplicate dot appears after "through sophisticated attack strategies."
- In the second paragraph of Section 2, a space is missing after "against downstream vision-language tasks."
- Sections 3 and 4 could be merged into a single section.
- In Section 5.1, there is a space missing after "abstract domain computations."
- Add references to the five defense methods in Section 5.3.
- In Section 6, there is a misplaced space after "formal robustness guarantees are essential."
- The last sentence of Section 6 is too long.

**Questions:**
- How is the "WordNet distance with embedding cosine similarity" precisely defined, and what is the associated value range for $\epsilon_{\mathrm{sem}}$?
- How are the hyperparameters $\beta, \gamma$  of the perceptual distance $d_{\text{perc}}$ defined, and what is the associated value range for $\eta$?
- Is $\Phi$ in Eq. (14) the certified MLLM built on top of the raw MLLM?
- Do the reported results in Section 5 always refer to both tasks from Table 1?

**References:**
- [I] Wang, Yanting et al. "MMCert: Provable Defense against Adversarial Attacks to Multi-modal Models." IEEE/CVF Conference on Computer Vision and Pattern Recognition. 2024.
- [II] Wang, Zhangyun et al. "Learning Robust Vision-Language Models from Natural Latent Spaces." Conference on Neural Information Processing Systems. 2025.

**Audience:**

Yes

**Audience Explanation:**

MLLMs, VLMs, and multimodal models in general are of high interest for the deep learning research community.

**Broader Impact Concerns:**

Currently, no broader impact statement is given. However, MLLMs have a large number of users worldwide. Therefore, strengthening the robustness of MLLMs is crucial. Providing a statement pointing out the importance of doing so, with exemplary benefits for society, would strengthen the paper's motivation. Moreover, the statement should also include potential damages resulting from the limitations of the proposed robustness framework.

**Claims And Evidence:**

Yes

**Claims Explanation:**

The empirical evaluation results and the computed certified output bound support all claims made by the authors. Due to my limited knowledge in this research area, I am unable to fully verify the correctness of these contributions.

**Requested Changes:**

- Provide more detailed step-by-step explanations in Section 4 and detail the mathematical notation as well as the values of the unclear hyperparameters.
- Extend your related work to consider related certified robustness works for multimodal models, e.g., [1].
- Provide your code either as supplementary material or via an anonymous link for review.
- Resolve the minor issues at the end of this review.
- Helpful but optional: Provide more detailed explanations regarding the computations of all performance metrics and attacks, e.g., in combination with equations as part of an appendix.

---

### Review · Reviewer_iysK · 2025-11-04

**Summary Of Contributions:**

This paper proposes a certified defense framework for MLLMs that provides formal robustness guarantees against coordinated cross-modal adversarial attacks. The core innovation is a hybrid polytope-zonotope abstraction that unifies two types of perturbations within a single mathematical framework: (1) polytopes exactly represent discrete semantic text substitutions, and (2) multi-norm zonotopes capture continuous image perturbations respecting perceptual metrics. The framework propagates these hybrid representations through all layers of multimodal transformer architectures using sound abstract transformers for operations like attention, linear transformations, and cross-modal fusion. Unlike existing empirical defenses that rely on adversarial training and finite test sets, this approach provides mathematical certification that holds for all possible perturbations.

**Strengths:**
- The problem addressed is important and of interest.
- The work seeks to provide provable robustness guarantees against joint cross-modal attacks, unlike empirical defenses that only test against finite attack samples.
- Hybrid polytope-zonotope abstraction unifies discrete text and continuous image perturbations.

**Weakness:**
- The paper is poorly written, containing grammatical errors, misaligned figures, and missing details.

**Audience:**

Yes

**Audience Explanation:**

Yes. Topic tackled in this work is important and timely. The paper seeks to offers both theoretical contributions and practical results that TMLR's audience would find valuable.

**Broader Impact Concerns:**

I have no concerns.

**Claims And Evidence:**

No

**Claims Explanation:**

The claims are difficult to verify because the presented evidence is often unclear or the details are missing. Specifically,

- The empirical results are unclear. For example, it is not clearly specified whether the “88.5% clean accuracy” is averaged across all four MLLMs and both datasets.

- The paper does not clearly specify whether or how the evaluated models were trained or fine-tuned. Given the substantial accuracy improvements over standard pretrained models (e.g., 49.1% → 74.6% on LPIPS-PGD), it appears that certified training may have been employed, but this is never explicitly stated. If certified training was employed, it should be clearly described and fairly compared against other training-based defenses; if not, the authors should clarify how such large improvements were achieved and how the evaluation was conducted.

- The paper doesn't discuss certification time or computational overhead. Computing these geometric abstractions through entire transformer models could be extremely expensive.

**Requested Changes:**

The paper addresses an important issue but requires clearer writing. I recommend the following improvements:

- Explain more clearly how you evaluate your method and how to interpret the results. Also, explain the baseline in more detail.

- Present the results separately for each of the four MLLMs and both datasets.

- Add more implementation details. For e.g., the values of several hyperparameters is missing.

- Discuss certification time or computational overhead.

- Give a clearer explanation of the evaluation metrics and ensure consistent use of their abbreviations; for example, Section 5.2 introduces “CA-Accuracy,” “AA,” and “CA-Cert,” but these terms are never reflected in the tables.

- Fix grammatical errors throughout the paper; e.g., missing periods and missing spaces after periods.

- Improve figure formatting; for e.g., in Figure 1, the arrows are misaligned with the boxes. The figure would also benefit from a more detailed caption.

---

### Review · Reviewer_JXZJ · 2025-12-02

**Summary Of Contributions:**

The paper presents a certified robustness method by propagating polytopes and zonotopes through an abstract model of transformer VLMs, and describes impressive empirical results.

**Audience:**

Yes

**Audience Explanation:**

The techniques and results referred to in this paper would be of great interest to the TMLR audience.

**Broader Impact Concerns:**

No broader impact concerns.

**Claims And Evidence:**

No

**Claims Explanation:**

The core contribution of the paper, sections 4.2, 4.3 and 4.4, is  not explained clearly. The perturbation domains are only gestured at (what does it mean to “combine” WordNet distance with embedding cosine distance? which embeddings, etc.?) The abstract transformer is a sketch of a plausible sounding method, but leaves out many extremely important details! The robustness bound seems to just be a bare assertion of correctness, and also refers to some expression \Phi which as far as I can tell is not defined anywhere.

It’s possible that some these techniques are all standard in this sub-area, then they should be cited and explained clearly. If they are new, then they need much much more explanation to explain how they work.

There’s also a missing detail which is left out or at least not apparent. A certifiable robustness technique like this can be applied to any choice of network weights (just, performance may be really bad if it’s not trained properly). Typically good performance requires some extra training. Was that the case here, or is it applied directly to a pretrained VLM?

Finally, I have multiple points of confusion about the experimental results. For example, certified accuracy and certification rates are reported for ProEAT, SafeMLLM, and MLLM-Protector but as far as I can tell these are empirical defenses, not certifiable, so I don’t understand what these results could mean. (This is just my immediate concern with the experiment section, there may be other problems.)

As this is a very interesting and important application, I welcome more exposition by the authors about their method, as well as correction of any misunderstandings I may have expressed here, and am open to changing my opinion if it is mistaken. However, as presently written, I think the concerns above are enough for me to conclude that the paper does not present clear and convincing evidence of the effectiveness of its technique.

**Requested Changes:**

See above: please give much more detail about the core methodology, and give more explanation of the experiment section.

---

### Decision · Action_Editor_Aod4 · 2026-01-08

**Recommendation:** Reject

**Audience:**

Yes

**Audience Explanation:**

There was general agreement that the problem and claimed results would be interesting to a part of the TMLR audience.

**Claims And Evidence:**

No

**Claims Explanation:**

All three reviews stated that they could not understand important information about the methods and evaluation, and gave this as the primary reason for a recommendation to reject.  Please see the individual reviews for a discussion of the gaps.  We invite the authors to submit a substantial revision that presents the methods and evaluation more precisely and fills in these gaps.

**Resubmission Of Major Revision:**

The authors may consider submitting a major revision at a later time.